



# Evaluation and climate sensitivity of the PlaSim v.17 Earth System Model coupled with ocean model components of different complexity

Michela Angeloni[1,2], Elisa Palazzi[2], and Jost von Hardenberg[3,2]

[1]Alma Mater Studiorum - University of Bologna, Department of Physics and Astronomy, Bologna, Italy
[2]Institute of Atmospheric Sciences and Climate, National Research Council (ISAC-CNR), Torino, Italy
[3]Politecnico di Torino, Department of Environment, Land and Infrastructure Engineering, Torino, Italy

**Correspondence:** M. Angeloni (m.angeloni@isac.cnr.it)

**Abstract.**

A set of experiments is performed with coupled atmosphere-ocean configurations of the Planet Simulator, an Earth-system Model of Intermediate Complexity (EMIC), in order to identify under which set of parameters the model output better agrees with observations and reanalyses of the present climate. Different model configurations are explored, in which the atmospheric

module of PlaSim is coupled with two possible ocean models, either a simple mixed-layer (ML) ocean with a diffusive transport parameterization or a more complex dynamical Large-Scale Geostrophic (LSG) ocean, together with a sea-ice module. In order to achieve a more realistic representation of present-day climate, we performed a preliminary tuning of the oceanic horizontal diffusion coefficient for the ML ocean and of the vertical oceanic diffusion profile when using LSG. Model runs under present-day conditions are compared, in terms of surface air temperature, sea surface temperature, sea ice cover, precipitation, radiation fluxes, ocean circulation, with a reference climate from observations and reanalyses. Our results indicate

that, in all configurations, coupled PlaSim configurations are able to reproduce the main characteristics of the climate system, with the exception of the Southern Ocean region in the PlaSim-LSG model, where surface air and sea surface temperatures are warm-biased and sea ice cover is by consequence highly underestimated.

The resulting sets of tuned parameters are used to perform a series of model equilibrium climate sensitivity (ECS) experiments, with the aim to identify the main mechanisms contributing to differences between the different configurations and

leading to elevated values of ECS. In fact, high resulting global ECS values are found, positioned in the upper range of CMIP5 and recent CMIP6 estimates. Our analysis shows that a significant contribution to ECS is given by the sea-ice feedback mechanisms and by details of the parameterization of meridional oceanic heat transport. In particular, the configurations using a diffusive heat transport in the mixed layer present an important sensitivity in terms of radiative forcing to changes in sea-ice

cover, leading to an important contribution of sea-ice feedback mechanisms to ECS.



# 1   Introduction

While numerical Global Climate Models (GCMs) are continuously improved to understand the main processes at work in the climate system and to evaluate the impacts of changes in radiative forcing, the complexity of these models has considerably grown over the past decades and simpler models have remained attractive for addressing research questions, understanding specific processes at work and for academic education. In these models atmospheric and climate processes are simplified

or highly parameterized so that key mechanisms become easier to study. In particular, Earth-system Models of Intermediate Complexity (EMICs), are global models which include the majority of the climate system components (atmosphere, ocean, cryosphere, vegetation, land surface) and their interactions, but in a simplified and parameterized form. This approach gives the opportunity to achieve an overall reproduction of the climate system, but also to isolate fundamental processes and study them in detail. Furthermore, owing to their relatively coarse resolution, EMICs offer the advantage of reduced computation

time with respect to the more complex GCMs, with the possibility of simulating the climate system over long time scales and exploring a wide parameter space.

The Planet Simulator (PlaSim) is an EMIC developed at the University of Hamburg (Fraedrich et al., 2005a, b; Fraedrich, 2012) which has been used, among other applications, to explore past conditions of the Earth such as the snowball Earth (Micheels & Montenari, 2008) or the Permian climate (Roscher et al., 2011), to perform specific experiments to test processes

like tropical convection or mid-latitude storm tracks using an Aquaplanet configuration (Dahms et al., 2011, 2012; Hertwig et al., 2014), to investigate the global entropy budget (Fraedrich & Lunkeit, 2008), and for exoplanetary studies (Kilic et al., 2017). The source code of the model is freely available (https://www.mi.uni-hamburg.de/en/arbeitsgruppen/theoretische-meteorologie/modelle/plasim.html). The oceanic component of the model can be either represented by a simple mixed-layer model (ML) (Lunkeit et al., 2011) or by a more complex, though simplified, fully 3D dynamical ocean model, the Large Scale

Geostrophic (LSG) model, based on primitive equations (Maier-Reimer et al., 1993). While the LSG ocean module allows to use Plasim as a fully coupled Atmosphere-Ocean GCM, it has found only limited application so far, mainly in aquaplanet and paleoclimatic studies (Dahms et al., 2012; Hertwig et al., 2014; Andres & Tarasov, 2019). Other configurations have also been developed, such as an integration of Plasim with the oceanic component of GENIE (Holden et al., 2016).

While a tuning exercise for the atmospheric component of PlaSim has been performed recently (Lyu et al., 2018), a tuning

and assessment of the climatology of coupled configurations of Plasim is still missing and is desiderable in particular to use the model for climate change studies, assessing tipping point mechanisms and to explore processes relevant for climate sensitivity. In particular, an evaluation of model equilibrium climate sensitivity represents a useful test to investigate the representation of associated feedbacks in the model. To this end, in this paper we present a set of experiments performed with PlaSim coupled with both ML and LSG ocean models, aimed at identifying which model configurations and parameter sets allow to better

reproduce the characteristics of the present climate when the model outputs are compared to climatologies obtained from observations and reanalyses. The obtained model configurations are then used to calculate the equilibrium climate sensitivity of PlaSim, to be compared with the values obtained from state-of-the-art EMICs and GCMs, such as those belonging to the Coupled Model Intercomparison Project phase 5 and 6 (CMIP5 and CMIP6; IPCC (2013); Andrews et al. (2012); Zelinka et



al. (2020); Meehl et al. (2020)). Two processes which we find to have a significant impact on equilibrium climate sensitivity, sea-ice feedbacks and meridional oceanic heat transport, are investigated in more detail.

The paper is structured as follows: in Section 2 the various components of the PlaSim model, the characteristics of the ML and LSG ocean models and their coupling with the atmospheric component are described; Section 3 presents the tuning of ocean parameters in the PlaSim-ML and PlaSim-LSG configurations; in Section 4 the simulated climate is evaluated, including
an assessment of the energy and water balance of the model; Section 5 deals with the equilibrium climate sensitivity of the model in the selected configurations and Section 6 summarizes the main results and concludes the paper.

## 2  Model description

### 2.1  The Planet Simulator model with a mixed-layer ocean

In this section, the main characteristics of the PlaSim Earth system Model of Intermediate Complexity (Fraedrich, 2012) are
briefly illustrated, but we refer to the PlaSim reference manual, Lunkeit et al. (2011), for further details. The dynamical core of PlaSim is a simplified General Circulation Model, the Portable University Model of Atmosphere (PUMA), based on the moist primitive equations representing the conservation of momentum, mass and energy (Fraedrich et al., 2005a) and using spectral methods to numerically solve them (Orszag, 1970; Eliasen et al., 1970). In the vertical, a $\sigma$-coordinate system and a finite-difference method to solve equations are used. The equations are time integrated with a leap-frog semi-implicit time
stepping scheme with time filter (Hoskins & Simmons, 1975; Simmons et al., 1978; Robert, 1981; Asselin, 1972).

All subgrid unresolved processes, and their effects, are included by means of parameterizations: surface fluxes (Roeckner et al., 1992), oceanic vertical and horizontal diffusion (Roeckner et al., 1992), shortwave (Lacis & Hansen, 1974; Stephens, 1984) and longwave radiations (Sasamori, 1968) (Stephens et al., 1984), moist processes (Kuo, 1974), clouds (Slingo & Slingo, 1991), dry convection are among the parameterized processes in this model.
PlaSim has been developed and tested to be run mainly at a horizontal spectral resolution of T21 (triangular truncation at wavenumber 21) and T42 (wavenumber 42). The computation of physical processes is done on the corresponding 5.6° longitude-latitude and 2.8° lon-lat reduced Gaussian grids, respectively. PlaSim has typically 10 atmospheric layers in the vertical up to 40 hPa. Appropriate computational time steps are 45 minutes for T21 and 30 minutes for T42. Surface boundary condition data are provided from four different sources: the U.S. Geological Survey LSP dataset (Hagemann et al., 1999) and
GTOPO30 dataset (Tibaldi & Geleyn, 1981), the MODIS satellite data (Rechid et al., 2009) and the AMIP-II sea-ice cover and sea surface temperature dataset (Taylor et al., 2000) for present day simulations of climate.

Sea surface temperatures can be simulated using a mixed-layer (ML) ocean model (Lunkeit et al., 2011) with constant thickness (the default value is 50 m). This ocean model consists of a prognostic equation for the oceanic temperature at each ocean grid point, which depends on the net atmospheric heat flux into the ocean. Ocean transports can be represented by the
heat convergence at the base of the ML (a so-called Q-flux) derived from climatology, but since the Q-flux approach may not be





suitable for climate studies under conditions far from present day, we will not focus on this configuration in this study. Instead oceanic transports are parameterized by the addition of an horizontal diffusion term to the temperature equation:

$$\frac{\partial T_{ML}}{\partial t} = F_a + K_h \nabla^2 T_{ML} \tag{1}$$

where $T_{ML}$ is the mixed-layer temperature, $F_a$ describes the net energy exchanges with the atmosphere and $K_h$ is a horizontal temperature diffusion coefficient (with a starting default value of $1000\ \mathrm{m^2 s^{-1}}$, but we tuned this value in our experiments).

The sea-ice distribution can either be prescribed by climatology or simulated by a thermodynamic sea-ice model based on the zero layer model by Semtner (1976), which computes the thickness of sea ice from the thermodynamic balance at the top and at the bottom of the sea-ice layer. This model assumes a linear temperature gradient in the ice and prevents ice from storing heat. Sea ice is formed if the ocean temperature drops below the freezing point (set to 271.25 K) and is melted if the ocean temperature exceeds that value (Lunkeit et al., 2011).

## 10   2.2   The Large Scale Geostrophic ocean circulation model

The Large Scale Geostrophic (LSG) ocean circulation model (Maier-Reimer et al., 1993; Drijfhout et al., 1996) is based on the primitive equations in a three-dimensional system, including the momentum equation, the continuity equation describing conservation of water and salinity, the thermodynamic equation with salinity. Please see the Large Scale Geostrophic Model report, Maier-Reimer & Mikolajewicz (1992), for details. The model is based on the observation that, since for a large scale

ocean circulation model developed for climate studies the characteristic spatial scales are large compared with the internal Rossby radius of deformation and the characteristic temporal scales are large compared with the periods of gravity modes and barotropic Rossby wave modes (Hasselmann, 1982), the nonlinear terms in the Navier-Stokes equations can be neglected. Furthermore the vertical friction is neglected and the hydrostatic and the Boussinesq approximations are applied (Maier-Reimer & Mikolajewicz, 1992).

Turbulent motions are parameterized by means of a vertical oceanic diffusion coefficient, $A_v$, which is a rather simple function of the vertical coordinate, $z$ (Bryan & Lewis, 1979):

$$A_v(z) = a^* + a_{\mathrm{range}} \arctan\left[\lambda(z - z^*)\right] \tag{2}$$

where $a^*$ is the vertical diffusion coefficient at a reference depth $z^*$, $a_{\mathrm{range}}$ defines the considered depth range from the surface to the bottom, and $\lambda$ is the rate at which the vertical diffusion coefficient varies with depth near $z^*$.

A long time step of 10 days is permitted by the implicit time integration scheme. The model has two staggered 5° x 5° horizontal grids (yelding an effective grid resolution of 3.5°), so that the variables of the model are defined on a semi-staggered E-type grid (Arakawa & Lamb, 1977). The components of horizontal velocity and the wind stress are defined on "vector points", while potential temperature, salinity, heat and freshwater fluxes, sea-surface height, pressure and vertical velocity are defined on "scalar points". The depth of the scalar points is usually defined as the maximum depth of the four surrounding vector points.





The w-points (for the vertical component of the velocity) are located between scalar points (Maier-Reimer & Mikolajewicz, 1992). By default, the number of oceanic vertical layers is 22, extending from the surface down to an oceanic depth of 6000 m. The Levitus-98 dataset (https://www.esrl.noaa.gov/psd/data/gridded/data.nodc.woa98.html) provides temperature and salinity initial conditions.

### 2.3    Coupling the PlaSim model with LSG

5    PlaSim and LSG are coupled through the surface fluxes of momentum, heat and fresh water. The atmospheric and oceanic grid interpolation ensures global conservation of energy and water (Fraedrich, 2012; Lorenz, 2006).

     The uppermost layer of the ocean regulates heat fluxes. The ML depth of PlaSim $\Delta z_{ML}^{(Pl)}$ and the upper layer thickness of LSG $\Delta z_{ML}^{(LSG)}$ are fixed to 50 m. Since the LSG free surface elevation ($\zeta$) is only 1 % of the LSG mixed-layer thickness (50 m), $\zeta$ can be neglected.

10    At the beginning of each LSG time step $\Delta t^{(LSG)}$ (10 days), the average over $\Delta t^{(LSG)}$ of the PlaSim ML temperature is imposed as the temperature of the ocean upper layer $T_{ul}^{\mathrm{LSG}}$,

$$T_{ul}^{(LSG)} = \overline{T_{ML}^{(Pl)}} = \left(\Delta t^{(LSG)}\right)^{-1} \int\limits_{\Delta t^{(LSG)}} T_{ML}^{(Pl)} dt^{(Pl)} \tag{3}$$

     A full ocean step is performed in which LSG calculates the ocean heat flux due to advective (advection, horizontal diffusion) and convective (vertical transport, vertical diffusion, convective adjustments) processes. The resulting distribution of the upper 15 layer temperature after $\Delta t^{(LSG)}$ determines the ocean heat flux, which is then given back to PlaSim. Equation. 3 is further modified to take into account small differences in the ML depth used by PlaSim and the upper layer of LSG and to correctly close the energy balance (see Lorenz (2006) for details).

     Sea ice in LSG is prescribed as calculated in the ML module of PlaSim. When calculating over hundreds of years, sea ice grows unconstrained in some isolated grid-points in the Antarctic ocean. In order to avoid the consequent increase of salinity 20 of open water in the upper layer of LSG, sea-ice thickness is forced not to exceed 9 m.

     Furthermore, the atmospheric wind stress and the freshwater flux (with a constant annual mean flux correction) are averaged over the coupling interval before they are transferred to the ocean.

### 3    Coupled ocean tuning

     In the following we focus on the tuning of two oceanic parameters which we have found to be crucial in determining the ability 25 of the model in reproducing the observed reference climate. In particular we consider: i) the horizontal diffusion coefficient (when using the ML ocean) and ii) the vertical diffusion coefficient (when using the LSG ocean).

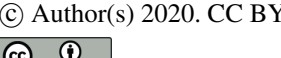



### 3.1 Mixed-layer ocean horizontal diffusion

The simulations with PlaSim-ML were performed including the dynamic sea-ice module, at T21 spatial resolution and with a fixed atmospheric $CO_2$ concentration (perpetual simulations) set to 354 ppm (representing conditions in 1990). Please notice that in the following we are comparing results from simulations with fixed $CO_2$ concentrations with observed climatologies in the years 2005-2015: using $CO_2$ concentrations from a previous period (1990) aims at compensating climate warming experienced by a climate model under fixed greenhouse gas forcing, assuming a feedback parameter of about 1 $\mathrm{Wm^{-2}K^{-1}}$ and the current radiative imbalance at TOA of about 0.5 $\mathrm{Wm^{-2}}$. We performed six 60 year long runs, with fixed atmospheric $CO_2$ concentrations, each having a different horizontal diffusion coefficient, $K_h$, which was varied in the range from $10^3\mathrm{m^2s^{-1}}$ to $10^6\mathrm{m^2s^{-1}}$. Panel $a$ of Fig. 1 shows the global mean of surface air temperature time series for all these simulations, each colored line representing one simulation performed with a different value of $K_h$.

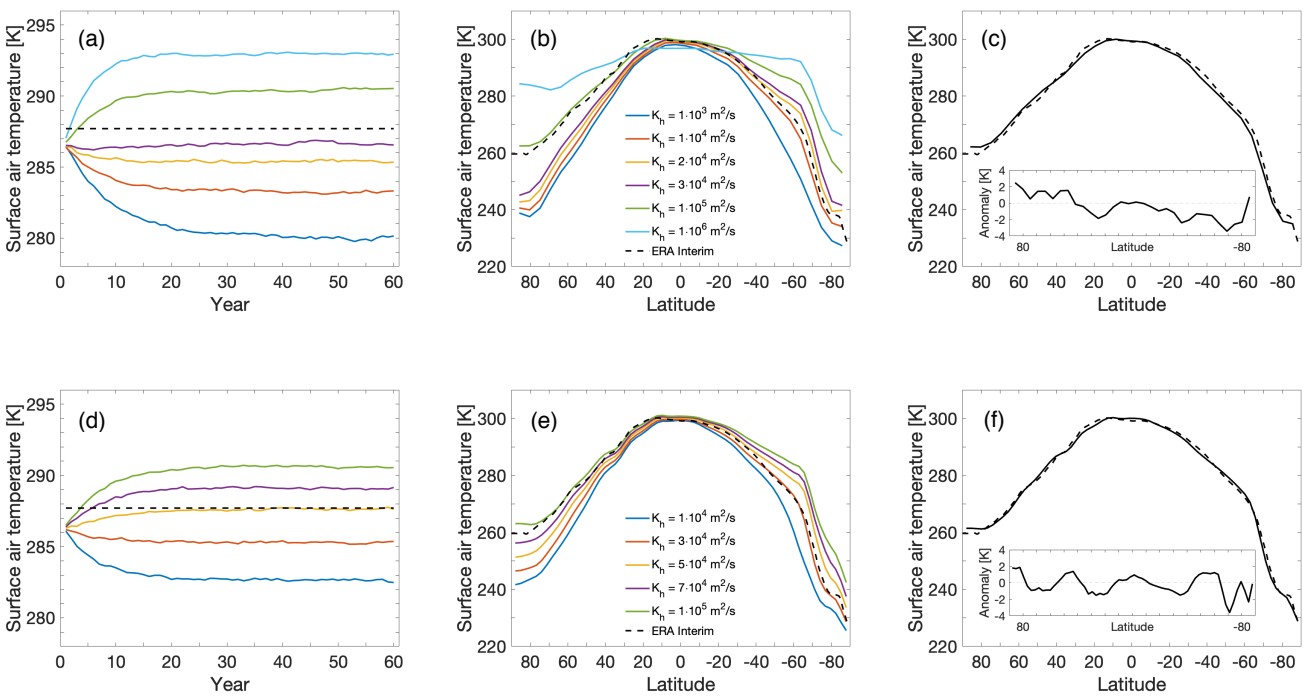

**Figure 1.** Time series and zonal mean of surface air temperature for different horizontal diffusion coefficient in the mixed layer, with the T21 (top) and T42 resolution (bottom). Right panels $c$ and $f$ show zonal mean of surface air temperatures (and anomalies with respect to ERA Interim in the inner boxes) for the simulations performed with two different coefficient in the Northern and the Southern Hemisphere. The ERA Interim values are the dashed lines.





10   As shown in the figure, the model in this configuration reaches an equilibrium state after about 30 years of simulation. The dashed black line in panel $a$ shows the global mean over the period 2005-2015 of near-surface air temperature from the ERA-Interim reanalysis dataset (Dee et al., 2011), to be compared with the model outputs.

In order to identify which $K_h$ value(s) leads to a better agreement between the model and the observations, we first computed, for each perpetual simulation, a time average over the last 30 years and then we compared the zonally-averaged surface air

temperature in the model and in the ERA-Interim dataset, as shown in Fig. 1 panel $b$. The figure clearly suggests that by choosing a single value of $K_h$ for the entire globe leads to either a cold bias in the Northern Hemisphere (NH) or a warm bias in the Southern Hemisphere (SH). In particular, the zonal mean temperatures in the simulation performed with $K_h = 10^5 \mathrm{m^2 s^{-1}}$ (green line) are in a good agreement with the reanalysis data in the NH, but give rise to a warm bias (up to 21 K) in the SH. Using instead $K_h = 10^4 \mathrm{m^2 s^{-1}}$ (red line), simulated surface air temperatures are in good agreement with the reanalyses in

the SH but they are too cold (up to 20 K) in the NH. This suggests the use of two different diffusion coefficients, to be separately applied in the Northern and Southern Hemisphere, i.e. $K_h = 10^5 \mathrm{m^2 s^{-1}}$ in the NH and $K_h = 10^4 \mathrm{m^2 s^{-1}}$ in the SH (this choice can be, to some extent, justified physically by the observed differences in meridional heat transport between the two hemispheres, particularly the strong North-South asymmetry observed in the Atlantic basin). Panel $c$ of Fig. 1 shows the results of this simulation, giving rise to a simulated zonally-averaged surface air temperature in very good agreement with the

reanalyses. The maximum difference between the model results and the ERA-Interim values is about 3 K.

We performed the same experiments (PlaSim-ML with dynamic sea ice, $CO_2$ = 354 ppm) running the model at higher resolution, T42. In this case, the horizontal diffusion coefficients $K_h$ has been tuned testing five different values, ranging from $10^4 \mathrm{m^2 s^{-1}}$ to $10^5 \mathrm{m^2 s^{-1}}$. In Fig. 1, panel $d$ and $e$ show the results of these experiments in terms of, respectively, surface air temperature time series and zonally-averaged surface air temperature for each simulation performed with a different $K_h$ value.

Also in this case (T42 resolution) a single diffusion coefficient does not allow to find a good agreement between the model and the reanalyses at the global level. In particular, panel $e$ would suggest to use $K_h = 10^5 \mathrm{m^2 s^{-1}}$ (green line) in the NH and $K_h = 3 \cdot 10^4 \mathrm{m^2 s^{-1}}$ (red line) in the SH. The results of the simulation performed with the two coefficients are shown in panel $f$.

## 3.2   Large Scale Geostrophic ocean vertical diffusion

We performed the series of experiments with PlaSim coupled with the Large Scale Geostrophic ocean model using dynamic sea ice, a fixed atmospheric $CO_2$ concentration equal to 354 ppm and atmospheric resolution T21. We initially performed two perpetual, 2000 year long runs, each having a different vertical diffusion profile. For the first run (run 1), we used the vertical diffusion coefficient profile suggested by Bryan & Lewis (1979) (see Eq. 2) with a surface value of $A_v = 0.3 \cdot 10^{-4} \mathrm{m^2 s^{-1}}$ and a bottom value of $A_v = 1.3 \cdot 10^{-4} \mathrm{m^2 s^{-1}}$ (see Table 1 and Fig. 2, blue line). For the second run (run 2), we used modified param-

eters which were found by Sciascia (2008) in ocean-only tuning experiments, with a surface value of $A_v = 0.8 \cdot 10^{-4} \mathrm{m^2 s^{-1}}$ and a bottom value of $A_v = 1.3 \cdot 10^{-4} \mathrm{m^2 s^{-1}}$ (Table 1 and Fig. 2, magenta line).



**Table 1.** Vertical diffusion parameters used in LSG model simulations. $A_v$ n. 1, 2 and 3 correspond respectively to the blue, magenta and green profiles shown in Fig. 2.

| $A_v$ | $a^*[\mathrm{m^2s^{-1}}]$ | $a_{range}[\mathrm{m^2s^{-1}}]$ | $z^*[\mathrm{m}]$ | $\lambda[\mathrm{m^{-1}}]$ |
|---|---|---|---|---|
| run 1 | $0.7958 \cdot 10^{-4}$ | $0.3345 \cdot 10^{-4}$ | 2500 | $4.5 \cdot 10^{-3}$ |
| run 2 | $1.0479 \cdot 10^{-4}$ | $0.1673 \cdot 10^{-4}$ | 2500 | $4.5 \cdot 10^{-3}$ |
| run 3 | $0.8714 \cdot 10^{-4}$ | $0.2843 \cdot 10^{-4}$ | 2500 | $4.5 \cdot 10^{-3}$ |

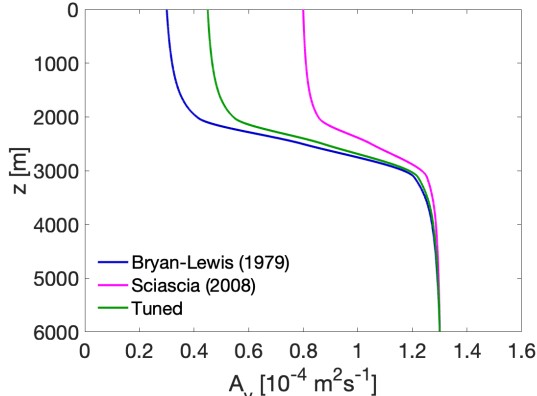

**Figure 2.** Vertical diffusion profiles for the LSG ocean model: $A_v$ suggested by Bryan & Lewis (1979) (blue line), $A_v$ suggested by Sciascia (2008) (magenta line) and the tuned $A_v$ profile (green line).

We extracted from the model outputs the time series of the maximum Atlantic Meridional Overturning Circulation (North Atlantic Deep Water) between 46-66° N and below 700 m (panel $a$ of Fig. 3). The thermohaline circulation collapses after about 500 years using coefficient n. 1 (blue line in Fig. 3 panel $a$), while it is active using coefficient n. 2 (magenta line), with values fluctuating from 17 to 27 Sv. The corresponding surface air temperature time series (panel $b$) show that the coupled model PlaSim-LSG reaches an equilibrium state after about 1000 years of simulation, and the simulated annual global mean surface temperatures are not in good agreement with the mean temperature from ERA Interim reanalysis data (dashed line) using these two coefficients. We computed the time mean over the last 1000 years and we analysed the zonal mean of surface air temperature (panel $c$) and the anomaly with respect to the ERA Interim dataset (inset box). In the Northern Hemisphere, the simulated temperature is negatively biased with coefficient n. 1 due to the AMOC collapse, while using coefficient n. 2 the PlaSim-LSG model maintains the oceanic circulation but overestimates (up to 12 K) surface air temperatures from 40° to 90° in the Southern Hemisphere.

In order to explore the contribution of the vertical diffusion at different depths, we performed simulations changing the surface and/or the bottom value of $A_v$ (not shown). This analysis revealed that the value of $A_v$ in the first 2000 m of the ocean



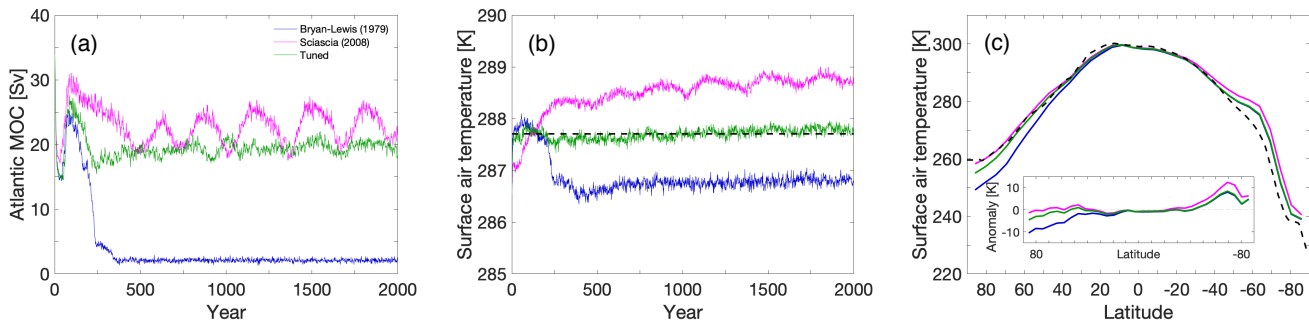

**Figure 3.** Maximum of the Atlantic Meridional Overturning Circulation (North Atlantic Deep Water) between 46-66° N and below 700 m (panel $a$). Time series (panel $b$) and zonal mean (panel $c$) of surface air temperature (and anomalies with respect to ERA Interim in the inner box) for different vertical diffusion profiles in the LSG ocean. The ERA Interim values are the dashed lines.

plays an important role, while its variation below 2000 m has no significant impact. Based on these results, we chose a vertical diffusion profile (run 3) with a surface value of $A_v = 0.45 \cdot 10^{-4} \mathrm{m^2 s^{-1}}$ and a bottom value of $A_v = 1.3 \cdot 10^{-4} \mathrm{m^2 s^{-1}}$ (see Table 1 and Fig. 2, green line), which is the best compromise between active AMOC (about 19 Sv) and lower temperature in the Southern Hemisphere (see Fig. 3). In this configuration, the coupled model PlaSim-LSG reaches an equilibrium state after about 500 years of simulation and the zonally-averaged maximum warm bias in the Southern Ocean is 8 K. In the attempt

to reduce this bias, we have modified other parameters (within physically acceptable limits), such as the cloud albedo, the continental ice-sheet albedo, the oceanic albedo and horizontal diffusion, the ozone concentration in the atmosphere, but with a negligible improvement of the resulting climate. Based on these results, we concluded that the simulation with profile n. 3 (green profile in Fig. 2) best reproduces temperature estimates from reanalyses and we chose it for the following PlaSim-LSG reference run.

**4   Model climate**

The preliminary tuning of ocean parameters described in the previous section allowed us to define three reference configurations which in the following we use to perform equilibrium climate sensitivity calculations. The first configuration consists of PlaSim coupled with the mixed-layer ocean, run at T21 spatial resolution and with two different horizontal oceanic diffusion coefficients, for the Northern Hemisphere ($K_h = 10^5 \mathrm{m^2 s^{-1}}$) and for the Southern Hemisphere ($K_h = 10^4 \mathrm{m^2 s^{-1}}$). The second

configuration is similar to the first one but the spatial resolution is finer (T42) and with a different horizontal diffusion oceanic coefficient in the Southern Hemisphere ($K_h = 3 \cdot 10^4 \mathrm{m^2 s^{-1}}$). In the third model set-up PlaSim is dynamically coupled with LSG and the vertical diffusion coefficient $A_v$ in the ocean is described by the function plotted in Fig. 2 (green line) and spans from $1.3 \cdot 10^{-4} \mathrm{m^2 s^{-1}}$ at the ocean bottom to $0.45 \cdot 10^{-4} \mathrm{m^2 s^{-1}}$ at the ocean surface. The characteristics of the mean climate under these configurations have been explored and are presented in this section.



## 4.1 Simulated climate

Figure 4 compares surface air temperature, sea surface temperature, sea-ice cover, precipitation and top-of-atmosphere (TOA) net radiation anomalies of PlaSim-ML and PlaSim-LSG against satellite observations and the ERA Interim reanalysis dataset, as summarized in Table 2. In particular the average over the last 30 years of the perennial simulation for PlaSim-ML and over the last 1000 years for PlaSim-LSG was compared with the average over the period 2005-2015 for the observational and reanalysis datasets.

**Table 2.** Observational and reanalysis datasets used in this paper.

| Variable | Dataset | Period | Horizontal resolution [°] |
|---|---|---|---|
| Surface air temperature | ERA-Interim (Dee et al., 2011) | 2005-2015 | 0.75° |
| Sea surface temperature | HadISST (Titchner et al., 2014) | 2005-2015 | 1° |
| Sea-ice cover | HadISST (Titchner et al., 2014) | 2005-2015 | 1° |
| Precipitation | GPCP (Adler et al., 2003) | 2005-2015 | 2.5° |
| Radiation | CERES-EBAF (Loeb et al., 2018) | 2005-2015 | 1° |
| Atlantic Meridional Overturning Circulation | Literature | - | - |

In general, simulated surface air temperatures are warm biased over the land pixels and cold biased over the ocean pixels. Surface air temperatures simulated with PlaSim-ML at T21 (panel $a$) are warm biased in Canada and Greenland and cold biased in the Barents Sea, while Antarctica shows both cold and warm anomalies. These anomalies are smaller using the T42 resolution (panel $b$). With PlaSim-LSG (panel $c$) the cold anomaly over the Barents Sea increases and a large warm bias (up to 20 K in some pixels) over the Southern Ocean is a very clear feature in this simulation. A similar pattern is observed in sea surface temperature anomalies (panels $d$-$f$), which show too warm temperatures on the western coast of America and Africa. Temperature anomalies in the polar regions are consistent with the simulated sea-ice cover. Using PlaSim-ML at T21 (panel $g$), the model simulates too little sea ice over the Arctic Ocean and too much sea ice over most of the Southern Ocean. With PlaSim-ML at T42 (panel $h$) sea ice is underestimated in both hemispheres but it is overestimated in the Barents Sea. Finally, PlaSim-LSG (panel $i$) leads to a strong negative sea-ice anomaly in the Southern Ocean, where sea ice is almost completely absent due to the warm bias. The simulated precipitation has a relatively good agreement with the observational data, except for the positive anomaly (about 10 mm/day) in the equatorial region using the model with the ML ocean (panels $l$ and $m$). PlaSim-LSG (panel $n$) better correlates with the observed precipitation than PlaSim-ML. Finally, TOA net fluxes (panels $o$-$q$) show a negative anomaly in the tropics and subtropics, where the upward radiation is overestimated, and a positive anomaly in the mid-latitude and polar zones, where the upward radiation is underestimated. Furthermore, a positive anomaly of net radiation is observed on the western coast of America and Africa, consistently with the warm biased surface air temperatures.





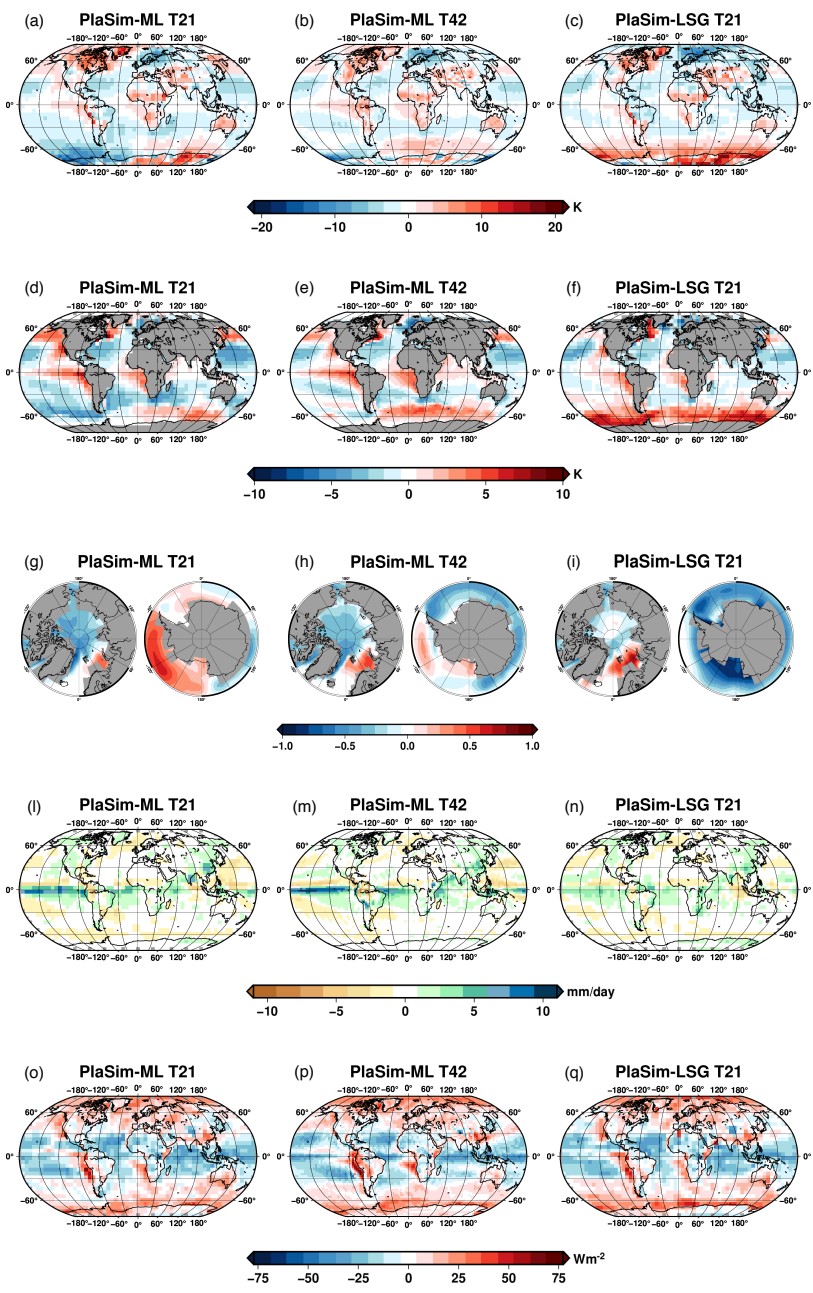

**Figure 4.** Difference between simulated and observed surface air temperature (panels $a$-$c$), sea surface temperature (panels $d$-$f$), sea-ice cover (panels $g$-$i$), precipitation (panels $l$-$n$) and TOA net radiation (panels $o$-$q$) for the three selected configurations of the model: PlaSim with the mixed-layer ocean and T21 (left), PlaSim with the mixed-layer ocean and T42 (middle), PlaSim with the LSG ocean and T21 (right).





## 4.2 Energy balance

For any coupled climate model to reach a stationary stable state, the TOA and the surface energy balances should be close to zero (neglecting geothermal heating) since the model should present no significant internal energy sources or sinks. Table 3 compares the global energy balance resulting from our reference simulations with estimates from Stephens et al. (2012). While the former are the results of perennial model simulations with PlaSim, the latter refers to an observed climatology calculated over the time period 2000-2010 during a period of climate change, so it presents positive TOA and surface radiative net fluxes. The last row of Table 3 shows the difference between the TOA and the surface net fluxes (which should be zero on average), indicating that none of the three PlaSim configurations conserves energy perfectly in the atmosphere. PlaSim-ML (with T21 resolution) and PlaSim-LSG configurations provide a negative balance indicating that the model atmosphere presents an internal energy source corresponding to 0.24 and 0.20 $\mathrm{Wm}^{-2}$, respectively. The PlaSim-ML (T42) configuration on the other hand, gives rise to a positive balance (there is a consumption of energy, corresponding to 0.12 $\mathrm{Wm}^{-2}$, in the model atmosphere). We tested if this imbalance is caused by a missing conservation of water mass in the model atmosphere (possibly to transport errors), but the absolute value of the global average freshwater flux P-E is smaller than $10^{-4}\mathrm{mm/day}$, equivalent to a very small latent heat flux smaller than $2.6 \cdot 10^{-3}\mathrm{Wm}^{-2}$ for all tested PlaSim configurations, indicating that water is well conserved in the PlaSim atmosphere. Also, all ML simulations present a negative net energy flux at the surface, suggesting some non-conservation of energy (equivalent to an energy production) in the mixed-layer ocean. Both the TOA-surface imbalance and the net surface flux bias are reduced in the T42 ML simulation, suggesting that these biases may be resolution-dependent. Overall these energy imbalances are small compared to those reported for CMIP5 and CMIP3 models (which could exceed 1 W/m$^2$ in magnitude at TOA (Mauritsen et al., 2012)).

**Table 3.** TOA and surface energy fluxes in $\mathrm{Wm}^{-2}$. Latent heat flux also includes the snow contribution.

|  | PlaSim-ML T21 | PlaSim-ML T42 | PlaSim-LSG T21 | Stephens et al. (2012) |
|---|---|---|---|---|
| TOA net shortwave | 231.5 | 235.8 | 232.8 | 240.2 |
| TOA net longwave | -232.3 | -236.0 | -232.9 | -239.7 |
| TOA energy budget | -0.76 | -0.11 | -0.14 | 0.6 |
| Surface net shortwave | 163.2 | 169.4 | 164.1 | 165 |
| Surface net longwave | -62.8 | -62.4 | -63.0 | -52.4 |
| Sensible heat flux | -18.9 | -20.8 | -18.3 | -24 |
| Latent heat flux | -82.0 | -86.5 | -82.7 | -88 |
| Surface energy budget | -0.52 | -0.23 | 0.06 | 0.6 |
| TOA-surface net | -0.24 | 0.12 | -0.20 | 0 |
| P-E imbalance (as latent heat) | $-2.6 \cdot 10^{-3}$ | $-2.3 \cdot 10^{-3}$ | $-1.9 \cdot 10^{-3}$ | - |





# 5 Equilibrium climate sensitivity and the role of sea ice

Equilibrium climate sensitivity (ECS) is defined as the equilibrium change in global mean surface air temperature after an instantaneous doubling of atmospheric $CO_2$ relative to pre-industrial levels (IPCC, 2013). Climate sensitivity can be diagnosed following the approach by Gregory et al. (2004) and here we apply this method to PlaSim-ML and PlaSim-LSG simulations. When a radiative forcing $R$ ($\mathrm{Wm^{-2}}$) is applied to the model, the model responds with a change in the net TOA radiative flux

$\Delta F$ ($\mathrm{Wm^{-2}}$) and, in order to restore the radiative equilibrium, the global mean surface air temperature, $\Delta T$, changes, until $\Delta F$ is returned to zero. $R$, $\Delta F$ and $\Delta T$ are related by the following equation:

$$\Delta F = R - \lambda \Delta T \tag{4}$$

where $\lambda$ ($\mathrm{Wm^{-2}K^{-1}}$) is referred to as climate feedback parameter. If $\Delta F$ is assumed to be a linear function of $\Delta T$, both the radiative forcing and the feedback parameter can be diagnosed by linear regression: $R$ is the intercept at $\Delta T = 0$ and $\lambda$ is the

slope (multiplied by -1). The equilibrium temperature change can be estimated extrapolating the heat balance to equilibrium, that is $\Delta F = 0$ and $\Delta T^{eq} = R/\lambda$. If the forcing is a doubling of $CO_2$, $\Delta T^{eq}$ is the equilibrium climate sensitivity by definition.

We performed a first set of simulations using dynamic sea ice (subscript $d$ in subsequent text): the first part of each simulation is a perennial run with pre-industrial boundary conditions, so the $CO_2$ concentration in the atmosphere is set to 285 ppm (1x$CO_2$); the second part is a perennial run in which the $CO_2$ concentration is instantaneously increased at 1.5, 2, 3 or 4 times

the value of the pre-industrial simulation. These simulations were made with the three reference configurations of PlaSim. Each half-simulation is 100 years long when using PlaSim-ML and 2000 years long when using PlaSim-LSG. The yellow and the red lines in Fig. 5 show the change in net TOA radiative flux versus the change in global mean surface air temperature for each dynamic-ice simulation with doubled (2x$CO_2$) and quadrupled (4x$CO_2$) atmospheric $CO_2$. Changes are computed with respect to the corresponding 1x$CO_2$ part of the simulation.

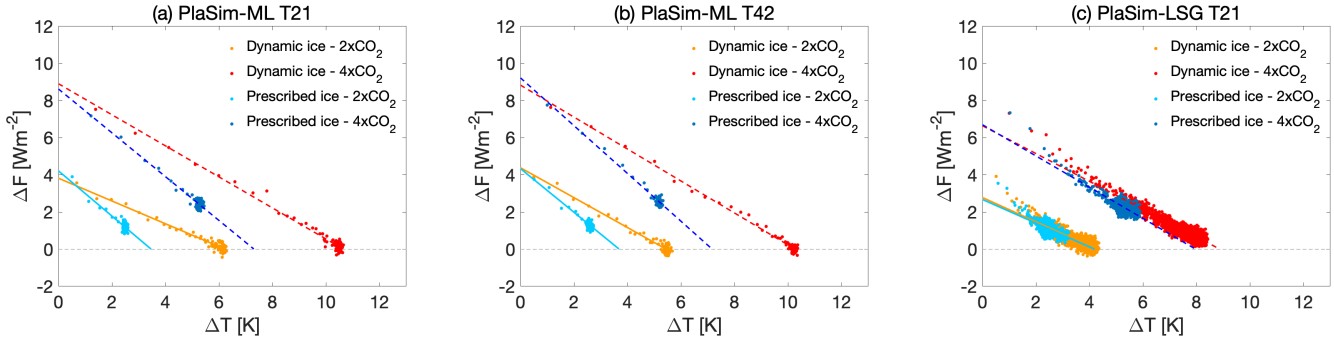

**Figure 5.** Relationships between $\Delta F$, the change in net TOA radiative flux, and $\Delta T$, the change in global mean surface air temperature, after an instantaneous doubling and quadrupling of $CO_2$, using dynamic or prescribed sea ice. Data points are global and annual means simulated with PlaSim-ML T21 (panel $a$), PlaSim-ML T42 (panel $b$) and PlaSim-LSG T21 (panel $c$). Lines represent ordinary least squares regression fits.



**Table 4.** Key values derived from simulations performed using the tuned configurations of PlaSim and four different $CO_2$ level increase factor: climate feedback parameter $\lambda_d$, radiative forcing $R_d$, equilibrium temperature $\Delta T_d^{eq}$, pre-industrial ($I_d^{PI}$) and final ($I_d$) sea-ice area at equilibrium for the dynamic-ice simulations; climate feedback parameter $\lambda_p$, radiative forcing $R_p$, equilibrium temperature $\Delta T_p^{eq}$ and sea-ice area for the prescribed-ice simulations; sea-ice feedback parameter $\lambda_i$, computed as the difference between $\lambda_d$ and $\lambda_p$. The reported uncertainties are standard deviations.

| PlaSim config. | $CO_2$ factor | $\lambda_d$ [$\mathrm{Wm^{-2}K^{-1}}$] | $R_d$ [$\mathrm{Wm^{-2}}$] | $\Delta T_d^{eq}$ [K] | $I_d^{PI}$ [$10^{12}\,\mathrm{m^2}$] | $I_d$ [$10^{12}\,\mathrm{m^2}$] | $\lambda_p$ [$\mathrm{Wm^{-2}K^{-1}}$] | $R_p$ [$\mathrm{Wm^{-2}}$] | $\Delta T_p^{eq}$ [K] | $I_p$ [$10^{12}\,\mathrm{m^2}$] | $\lambda_i = \lambda_d - \lambda_p$ [$\mathrm{Wm^{-2}K^{-1}}$] |
|---|---|---|---|---|---|---|---|---|---|---|---|
| ML T21 | 1.5 | $0.58 \pm 0.16$ | $2.10 \pm 0.55$ | $3.64 \pm 0.20$ | 29.81 | 15.64 | $1.08 \pm 0.62$ | $2.22 \pm 0.85$ | $2.05 \pm 0.40$ | 29.65 | $-0.50 \pm 0.64$ |
|  | 2 | $0.61 \pm 0.10$ | $3.82 \pm 0.56$ | $\mathbf{6.23 \pm 0.19}$ | 29.81 | 8.40 | $1.22 \pm 0.36$ | $4.22 \pm 0.89$ | $3.46 \pm 0.32$ | 29.65 | $-0.60 \pm 0.38$ |
|  | 3 | $0.71 \pm 0.07$ | $6.47 \pm 0.59$ | $9.16 \pm 0.16$ | 29.81 | 3.24 | $1.20 \pm 0.22$ | $6.87 \pm 0.91$ | $5.73 \pm 0.33$ | 29.65 | $-0.49 \pm 0.23$ |
|  | 4 | $0.84 \pm 0.06$ | $8.91 \pm 0.64$ | $10.65 \pm 0.13$ | 29.81 | 1.99 | $1.18 \pm 0.17$ | $8.62 \pm 0.90$ | $7.31 \pm 0.33$ | 29.65 | $-0.34 \pm 0.19$ |
| ML T42 | 1.5 | $0.79 \pm 0.19$ | $2.55 \pm 0.59$ | $3.23 \pm 0.13$ | 18.15 | 8.61 | $1.18 \pm 0.56$ | $2.55 \pm 0.83$ | $2.16 \pm 0.32$ | 18.06 | $-0.39 \pm 0.59$ |
|  | 2 | $0.81 \pm 0.11$ | $4.39 \pm 0.60$ | $\mathbf{5.45 \pm 0.13}$ | 18.15 | 4.12 | $1.18 \pm 0.33$ | $4.34 \pm 0.85$ | $3.69 \pm 0.34$ | 18.06 | $-0.37 \pm 0.35$ |
|  | 3 | $0.81 \pm 0.07$ | $6.90 \pm 0.61$ | $8.51 \pm 0.13$ | 18.15 | 0.80 | $1.17 \pm 0.21$ | $7.13 \pm 0.87$ | $6.08 \pm 0.34$ | 18.06 | $-0.36 \pm 0.22$ |
|  | 4 | $0.86 \pm 0.06$ | $8.83 \pm 0.62$ | $10.26 \pm 0.12$ | 18.15 | 0.39 | $1.29 \pm 0.18$ | $9.22 \pm 0.90$ | $7.17 \pm 0.30$ | 18.06 | $-0.43 \pm 0.19$ |
| LSG T21 | 1.5 | $0.59 \pm 0.07$ | $1.58 \pm 0.16$ | $2.68 \pm 0.06$ | 11.42 | 6.92 | $0.49 \pm 0.11$ | $1.32 \pm 0.15$ | $2.71 \pm 0.28$ | 11.44 | $0.10 \pm 0.13$ |
|  | 2 | $0.65 \pm 0.05$ | $2.78 \pm 0.17$ | $\mathbf{4.26 \pm 0.06}$ | 11.42 | 5.86 | $0.64 \pm 0.11$ | $2.68 \pm 0.27$ | $4.19 \pm 0.31$ | 11.44 | $0.02 \pm 0.12$ |
|  | 3 | $0.70 \pm 0.03$ | $4.96 \pm 0.19$ | $7.07 \pm 0.06$ | 11.42 | 3.55 | $0.77 \pm 0.09$ | $4.91 \pm 0.36$ | $6.35 \pm 0.27$ | 11.44 | $-0.07 \pm 0.10$ |
|  | 4 | $0.75 \pm 0.03$ | $6.64 \pm 0.20$ | $8.88 \pm 0.06$ | 11.42 | 1.58 | $0.84 \pm 0.07$ | $6.70 \pm 0.39$ | $7.97 \pm 0.25$ | 11.44 | $-0.09 \pm 0.08$ |

We derive the estimates of $R_d$ (intercept) and $\lambda_d$ (slope multiplied by -1) through ordinary least squares regression and we compute $\Delta T_d^{eq} = R_d/\lambda_d$ for all the PlaSim configurations (Table 4). The confidence intervals are obtained as standard deviation on the parameter estimates and error propagation. The resulting ECS for dynamic sea ice is 6.23 K using PlaSim-ML T21, 5.45 K using PlaSim-ML T42 and 4.26 K using PlaSim-LSG T21, using the results from the $CO_2$ doubling experiments. In Fig. 6 we compare these results with values from other models. In particular, the grey boxplots give an indication of the

distribution of CMIP5 values (the whiskers extend to the highest and lowest data) discussed in Andrews et al. (2012). Radiative forcing and climate feedback values of our model are within the range estimated for CMIP5 models, but only the PlaSim-LSG coupled model gives an equilibrium climate sensitivity within the CMIP5 range (2.1-4.7 K), though close to the upper limit. The orange boxplot represents the ECS values found in other EMICs (Pfister & Stocker, 2017), which are in good agreement with CMIP5 models but have a wider range of values (1.5-5.5 K). Finally, the blue boxplot shows the most recent range of ECS

for CMIP6 models (1.8-5.6 K; Zelinka et al. (2020)). With respect to CMIP5, some components of CMIP6 models have been improved, for example low clouds and shallow convection are better represented (Voldoire et al., 2019) or a more advanced treatment of aerosol is included (Wyser et al., 2020), and stronger positive cloud feedbacks from decreasing extratropical low cloud coverage and albedo have contributed to increased ECS in some of them (Zelinka et al., 2020; Meehl et al., 2020). Since PlaSim does not include such a level of accuracy (for example, it has no parameterization for aerosol-cloud feedbacks), its

high climate sensitivity cannot be related to these processes. Our results can also be compared also with an ECS estimate for a modified PlaSim ML configuration at T21 in Ragone et al. (2016), where the very high value of 8.1 K was reported and



attributed to the removal of the diurnal and seasonal cycles in the model. Please notice also that, while EMIC ECS values reported in Fig. 6 were obtained, like for PlaSim, from $CO_2$ doubling experiments, the reported CMIP5 and CMIP6 results were obtained dividing by two the results from 4x$CO_2$ experiments. As shown in Table 4 for PlaSim and as also reported in Pfister & Stocker (2017) for other EMICS, the ECS values obtained from quadrupling experiments may be lower than those obtained from doubling experiments, although often used without distinction as estimates of ECS in the literature.

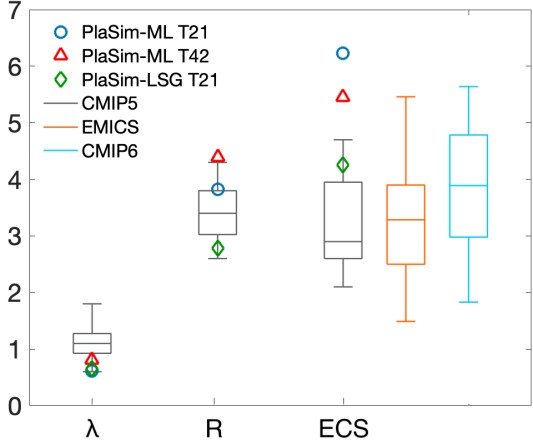

**Figure 6.** Radiative forcing $R$ (in $\mathrm{Wm^{-2}}$), climate feedback $\lambda$ (in $\mathrm{Wm^{-2}K^{-1}}$) and equilibrium climate sensitivity ECS (in K) values for PlaSim-ML (T21 and T42) and for PlaSim-LSG (T21). Boxplots show the corresponding ranges of values found in CMIP5 models (Andrews et al., 2012) and estimates of ECS values from EMICs (Pfister & Stocker, 2017) and CMIP6 models (Zelinka et al., 2020)

.

The difference between the two PlaSim-ML configurations and the PlaSim-LSG configuration can partly be explained by features of the ocean circulation. Using PlaSim-LSG, the AMOC is active (about 20 Sv) in the pre-industrial and 1.5x$CO_2$ runs, but it collapses (less than 5 Sv) in the 2x$CO_2$, 3x$CO_2$ and 4x$CO_2$ simulations. As shown in Fig. 3b, global average temperatures are affected significantly by the state of AMOC, with a cooling of up to 1°C in runs with a shutdown of AMOC. As a consequence, the equilibrium climate sensitivity in such runs is smaller than it would be if the AMOC had remained

active. Unlike PlaSim-LSG and CMIP models, the PlaSim-ML configurations doesn't include an AMOC representation, so it can't weaken and this could contribute to the reported higher ECS.

    The relatively high values of ECS found for PlaSim are related to low values of the feedback parameter $\lambda_d$. We determined that an important contribution can be traced also to elevated values in magnitude of the ice-feedback parameter, as we assessed following the approach of Caldeira & Cvijanovic (2014). To this end, we performed a second set of simulations, similar to

the first one but with prescribed sea ice (subscript $p$): twelve climatological monthly ice extents were derived from the pre-industrial dynamic-ice simulation and were prescribed in the model. The cyan and the blue lines in Fig. 5 show the change in net TOA radiative flux versus the change in global mean surface air temperature for each prescribed-ice simulation with doubled and quadrupled $CO_2$ concentration. Using prescribed sea ice, the change in TOA radiative flux at equilibrium is not





zero (see Fig. 5) because some energy has to be removed from or added to the system in order to maintain the climatological sea-ice thickness (Caldeira & Cvijanovic, 2014). Also in this case, we computed $\lambda_p$ (slope multiplied by -1) through ordinary linear least squares regression for all the PlaSim configurations (Table 4). The sea-ice feedback parameter $\lambda_i$ (last column of Table 4) is negative in our sign convention and is obtained subtracting the feedback parameter of dynamic-ice simulations from that of prescribed-ice simulations. These results can be compared with the slab-ocean experiments performed by Caldeira

& Cvijanovic (2014) with the National Center for Atmospheric Research's Community Earth System Model (CESM): they report a $\lambda_i$ of -0.21 $\pm$ 0.19 $\mathrm{Wm^{-2}K^{-1}}$ in the doubling $CO_2$ experiments and -0.30 $\pm$ 0.06 $\mathrm{Wm^{-2}K^{-1}}$ in the quadrupling $CO_2$ experiments. The feedback parameter of sea ice in the PlaSim-ML configurations is significantly higher in absolute value than in CESM, suggesting that sea ice plays an important role in determining the ECS of the model. However, this contribution also depends on the extent of sea ice either in the pre-industrial climate (see $I_d^{PI}$ in Table 4) or in the future climates (see

$I_d$). Indeed, the sea-ice area is very different in the three configurations of PlaSim. For example, in the PlaSim-ML model at T42 the pre-industrial sea-ice area is less extended than in the PlaSim-ML model at T21, so the sea-ice contribution to ECS is smaller. Furthermore, we recall that sea ice is almost completely absent in the Southern Ocean using PlaSim-LSG, therefore the sea-ice contribution to ECS is reduced compared to the configurations with the ML ocean. A factor which also contributes to the low values of $\lambda_i$ in the PlaSim-LSG configuration, is the fact that in our simulations in the pre-industrial simulation

with prescribed sea ice, the AMOC in the LSG ocean model collapses to very low values after about 1000 years, while using the dynamic sea-ice treatment the AMOC is strong in the pre-industrial climate. The AMOC collapse in the prescribed-ice simulation makes the pre-industrial global temperature colder (as reported above), the slope $\lambda_p$ smaller and the difference $\lambda_i$ closer to zero than it would be with an active AMOC. Therefore in the PlaSim-LSG model the feedback parameter $\lambda_i$, obtained subtracting dynamic-ice from prescribed-ice simulations, includes not only the sea-ice contribution but also the AMOC effect,

which is positive in our sign convention.

The high impact of sea-ice related feedbacks in the model cannot be linked to a too high sea-ice albedo in PlaSim: in fact we compared the average sea-ice albedo of PlaSim and that of EC-Earth, a global state-of-the-art climate model with higher complexity and spatial resolution (Hazeleger et al., 2011; Döscher et al., 2020), which has an ECS of 4.3 K in the newer model version (Wyser et al., 2020). The average sea-ice albedo of PlaSim-ML T21 is 0.58 for pixels with more than

99% area coverage, lower than the average sea-ice albedo of EC-Earth (0.80). Since a smaller sea-ice albedo weakens the ice-albedo feedback, we can conclude that the strong impact of dynamic sea ice in PlaSim is not likely due to the ice albedo parameterization employed in the model. In fact, we performed a series of climate sensitivity runs with PlaSim (not shown) in which we modified maximum sea-ice albedo and its dependence on temperature, without finding significantly lower values of the ECS.

A confusing factor between the different model configurations is the fact that they are all characterized by different average sea-ice extents in the starting preindustrial experiments. To better compare the impact of the sea-ice feedback, we can use





the same approach as Caldeira & Cvijanovic (2014), comparing the dynamic-ice and prescribed-ice simulations, to define a measure of the radiative forcing associated with changes in sea-ice area. An equivalent formulation with our symbols is:

$$\Delta F_{ice} = \lambda_p \left( \Delta T_d^{eq} - \Delta T_p^{eq} \right) \tag{5}$$

which represents a measure of the radiative forcing that should be provided to prescribed-ice experiments in order to undergo the same global mean surface air temperature change as in dynamic-ice experiments. Figure 7 shows the sea-ice radiative forcing versus the relative sea-ice area (computed as the difference between $I_p$ and $I_d$, see Table 4): the tuned configurations of PlaSim with ML and LSG (blue, red and green lines) are compared with CESM, used in Caldeira & Cvijanovic (2014) (black dashed line).

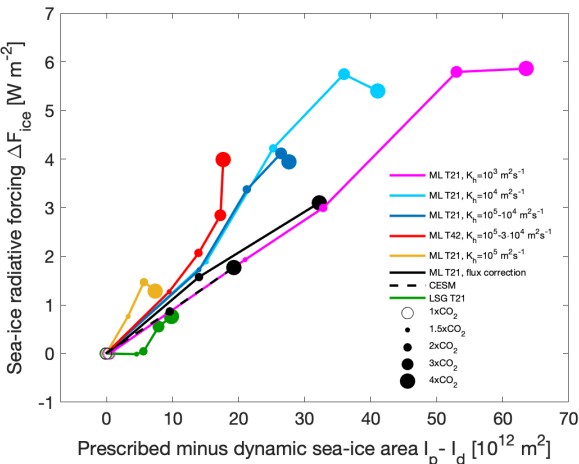

**Figure 7.** Sea-ice radiative forcing (see Eq. 5) as a function of the relative sea-ice area (prescribed-ice area minus dynamic-ice area, see Table 4). Values are obtained by estimating the radiative forcing that should be provided to prescribed-ice experiments in order to undergo the same global mean surface air temperature change as in dynamic-ice experiments.

The configurations of PlaSim with the mixed layer, in which the oceanic transport is parameterized by a horizontal diffusion term, show a radiative forcing associated with changes in sea-ice area which is significantly higher than CESM, consistently with the higher value of $\lambda_i$ in absolute value which we reported above. One important difference between our experimental results and the CESM results is that the latter experiments used a mixed-layer ocean with a prescribed ocean heat flux (Q-flux). In order to verify the role played by the specific parameterization of oceanic transport, we performed also a series of $CO_2$ increase experiments with dynamical and prescribed ice, using PlaSim with a Q-flux correction instead of horizontal diffusion in the mixed layer (the Q-flux was derived by the model from present-day AMIP experiment forced with observed SSTs). As shown in the figure, the configuration of PlaSim-ML T21 with the flux correction gives similar results to CESM in terms of ice forcing, suggesting that the specific choice of parameterization of oceanic heat transport affects the sea-ice radiative forcing





and as a consequence the ECS of the model. To further explore this point, Fig. 7 also shows three additional simulations which use a single horizontal diffusion coefficient for both the hemispheres, $K_h = 10^5 \mathrm{m}^2\mathrm{s}^{-1}$ (yellow line), $K_h = 10^4 \mathrm{m}^2\mathrm{s}^{-1}$ (cyan line) and $K_h = 10^3 \mathrm{m}^2\mathrm{s}^{-1}$ (magenta line). The first and the second coefficient are those used for the Northern and the Southern Hemispheres in the PlaSim-ML (T21) configuration. We can notice that PlaSim-LSG and PlaSim-ML with the lowest diffusion coefficient ($K_h = 10^3 \mathrm{m}^2\mathrm{s}^{-1}$) have a slope similar to CESM, while using a higher ocean diffusion, the effective changes
in radiative forcing associated with changes in sea-ice area increase for increasing horizontal oceanic diffusion. Therefore ultimately the equilibrium climate sensitivity of the PlaSim-ML model using a diffusive term to represent heat transport, depends crucially also on the choice of $K_h$.

## 6    Conclusions

In this paper the evaluation and the climate sensitivity of coupled configurations of the PlaSim model, using two different
ocean modules (a simple mixed layer or the Large Scale Geostrophic ocean circulation model), are presented. In order to parameterize heat transport in the ML, a horizontal diffusion coefficient $K_h$ was used, a configuration which potentially could be preferable for climate change or paleoclimatic studies, due to its ability to adapt heat transport to varying pole-equator gradients, compared to a fixed Q-flux.

     A preliminary tuning to achieve realistic zonally-averaged near-surface temperature profiles suggests the use of two different
values for $K_h$, one in the Northern Hemisphere (NH) and another one in the Southern Hemisphere (SH). To the same end, for the coupled PlaSim-LSG model we found that, when using the arctan-shaped vertical ocean diffusion profile following Bryan & Lewis (1979), values ranging from $0.45 \cdot 10^{-4} \mathrm{m}^2\mathrm{s}^{-1}$ at the top to $1.3 \cdot 10^{-4} \mathrm{m}^2\mathrm{s}^{-1}$ at the bottom of the ocean should best be used. Most climatic variables are well simulated and their anomalies are reduced using the finer T42 resolution. There is a warm bias in the Southern Hemisphere (between 40 and 90° S) and the Antarctic sea ice is underestimated in the coupled PlaSim-
LSG model. PlaSim, in all tested configurations, does not provide a perfect energy balance, probably due to its relatively coarse spatial resolution. In fact, we found smaller imbalance in the simulations at higher resolution. Still, overall these energy imbalances are small compared to those reported in literature for several other global climate models.

     The tuned PlaSim configurations were used to assess the equilibrium climate sensitivity (ECS) of the model from $CO_2$ doubling experiments with dynamic sea ice and to compare it with values found in CMIP5 models, other state-of-the-art
EMICs and CMIP6 models. The ECS of the model is found to be particularly high in the ML configurations, with an ECS of 6.23 K using PlaSim-ML at T21 and 5.45 K using PlaSim-ML at T42, compared to 4.26 K using PlaSim-LSG at T21. Only the latter value is within the CMIP5 range, though close to the upper limit. One important factor contributing to higher ECS in PlaSim-ML compared to PlaSim-LSG is that unlike the PlaSim-LSG configuration and complex CMIP models with a 3D ocean, in PlaSim with the ML ocean the AMOC can't weaken or collapse and we have seen also in our simulations that
reduced AMOC strength will have a cooling effect on the global average temperatures. The high ECS of the model in the ML configurations has also been found to be related to elevated values in magnitude of the ice-feedback parameter, as computed performing prescribed sea-ice simulations. In fact, overall, our climate sensitivity experiments with the PlaSim EMIC reveal

that details of the oceanic heat transport play a very important role in determining the sea-ice feedback parameter and as a consequence the ECS of the model. When using a diffusive term in the ML, with values of the horizontal diffusion parameter $K_h$ which allow for a realistic meridional temperature distribution in present-day experiments, changes in average sea-ice area have a much stronger radiative impact, compared to very low values of the diffusion coefficient or to using a Q-flux approach to represent transport. Since using a diffusive term may be preferable to a fixed Q-flux in some cases for studying climate

responses far from present-day conditions (such as paleoclimatic or eexoplanetary studies), this impact may have to be taken carefully into account. The fact that the PlaSim configuration with LSG presents a lower radiative impact of sea ice, comparable to that of experiments with a Q-flux, reveals that indeed sea-ice feedbacks may be overestimated in the configurations using a ML with a diffusive transport parameterization. An additional factor contributing to varying values of ECS between different configurations is that, due to the relevant impact of sea-ice feedbacks, the sea-ice extent in preindustrial simulations plays an

important role, and this contributes to the lower ECS of the configuration using LSG, which is affected by a strong southern-ocean bias and almost no sea ice in the southern hemisphere.

*Code and data availability.*  Original PlaSim distribution: https://github.com/HartmutBorth/PLASIM. All improvements described in this paper are included in this version of the code: https://github.com/jhardenberg/PLASIM (https://doi.org/10.5281/zenodo.4041462). The scripts and the data needed to reproduce the figures are available in the supplementary material.

*Author contributions.*  MA, EP and JvH developed the idea of the paper, discussed the analyses and wrote the manuscript; MA wrote the first manuscript draft, performed all simulations, analysed the outputs and prepared the figures.

*Competing interests.*  No competing interests are present.

*Acknowledgements.*  This project is TiPES contribution #40: This project has received funding from the European Union's Horizon 2020 research and innovation programme under grant agreement No. 820970 (TiPES).



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
