# Peer review of "Evaluation and climate sensitivity of the PlaSim v.17 Earth System Model coupled with ocean model components of different complexity"

_Geoscientific Model Development, 2020_

## Referee Comment (RC1) · Anonymous Referee #1 · 8 Jan 2021

The paper presents an evaluation of the PlaSim v.17 Earth System Model run with either a 50 m deep slab mixed layer ocean or coupled with a simple ocean model. The slab mixed layer configuration is tested at T21 and T42 atmospheric resolutions, whilst the configuration with the simple ocean model uses only T21. Having tested a small selection of values for the horizontal diffusion coefficient in the slab mixed layer an alternative is chosen whereby the northern and southern hemisphere are prescribed different values of the horizontal diffusion coefficient. This improves the fit with the observed time- and zonal mean surface air temperature. Three values (profiles) of the vertical diffusion coefficient are tested in the dynamic ocean. The equilibrium climate sensitivity of the preferred model configurations are tested, and are broadly found to

be near or above the upper bounds of those reported from CMIP5, other EMICS and even to CMIP6. The results are found to be particularly sensitive to the dynamic sea ice, and in the case of the slab mixed layer the constant value(s) of the horizontal diffusion coefficient do not simulate the expected weakening of the poleward ocean heat transport that would occur with a slowing of the Atlantic MOC.

Overall I find the experimental design of the tuning (Section 3) very underwhelming. The speed and simplicity of the model would make it easy to perform an extremely rigorous calibration, following examples such as Murphy et al. (2004), Stainforth et al. (2005), Williamson et al. (2013, 2015), Shi et al. (2019), to name but a few. In fact, as mentioned in the introduction, Lyu et al. (2018) demonstrated an adjoint-based tuning approach for PlaSim, so I wonder why the authors chose not to build upon this work? The tuning presented is no more than testing a small set of values for one parameter and selecting the one which has the best visual fit to the zonal average surface air temperature.

The evaluation of the modelled climate states (Section 4) is very brief and it does not contain enough analysis to convince the reader that a reasonable time- and zonal mean SAT is being achieved for the right reasons or without introducing significant biases in quantities not shown, such as poleward heat transports. No confidence intervals are presented (e.g. in Table 3 or Figure 4) to indicate whether the climate is near/within the observational range.

Existing studies that explore equilibrium climate sensitivity in GCMs and EMICs with dynamic and slab ocean configurations should be acknowledged (see e.g.: Danabasoglu and Gent, 2009; Stouffer and Manabe, 1999; Shell, 2013). Danabasoglu and Gent (2009) in particular discuss the roles of sea ice area and the slab ocean heat flux transport in a similar experimental setup, and there are some interesting points of similarity and difference with respect to this study that would be worth discussing. The role of oceanic heat transport in climate model sensitivity experiments was discussed in some detail in Spelman and Manabe (1984) using a coupled ocean-atmosphere model

and a mixed-layer ocean-atmosphere model, and again this literature is not cited or discussed.

Considering the above shortcomings, I think the study in its current form is of limited interest to readers, and it does not represent a substantial advance in modelling science. It could be improved sufficiently by greatly improving the model evaluation and the discussion of the results in context with existing literature.

Minor points:

The terms 'experiment' and 'simulation' are used synonymously in many places. This can be confusing (e.g. P14 L14, P15 L11).

P3 L18: All subgrid unresolved processes? On P13 L13 you state PlaSim "has no parameterization for aerosol-cloud feedbacks".

P3 L20 "(Sasamori, 1968) (Stephens et al., 1984)" should this be one set of brackets, or is there text missing in between?

P3 L26: Please state what surface boundary conditions each of these datasets provide. LSP undefined

P4 L15-18: This is a very long sentence. Please reword.

P4 L26: Two staggered grids? Please can you describe this further?

Figure 1: The caption is ambiguous. "simulations performed with two different coefficient in the Northern and the Southern Hemisphere" could be interpreted in several ways.

Table 1: According to https://www.metoffice.gov.uk/hadobs/hadisst/, the correct citation to use for HadISST is:

Rayner, N. A.; Parker, D. E.; Horton, E. B.; Folland, C. K.; Alexander, L. V.; Rowell, D. P.; Kent, E. C.; Kaplan, A. (2003) Global analyses of sea surface temperature, sea

ice, and night marine air temperature since the late nineteenth century J. Geophys. Res.Vol. 108, No. D14, 4407 10.1029/2002JD002670

P6 L13: Remove "." After Equation

P6 L17: By "isolated" do you mean not connected to the open ocean, or e.g. single grid point inlets?

Section 2: Please indicate the computational cost of the ML and LSG configurations of PlaSim.

What happens at the North Pole to prevent convergence of the meridians resulting in numerical instability?

P6 L20: What is the coupling interval? You mention the atmospheric time steps (P3 L25) and the ocean time step (P4 L26), but no details of the coupling time step.

P7 L17: This makes sense. Could you further justify this choice by computing the effective poleward heat transport due to the choice of diffusivity? Is this the first study to propose using different values of horizontal diffusivity in each hemisphere for a slab ocean model?

P9 L21: Please expand on how you explored parameter space. Did you only vary one parameter at a time?

Section 4.1 is very brief. Anomalies are described, but these need uncertainty metrics, and it would be nice to accompany this with some discussion of the causes.

Figure 4: I think this would work better as (at least) two figures. The labels on (g)-(i) are too small to read without zooming in considerably. Where are panels (j) and (k)? (a), (c), (d), (f), (l), (n), (o) and (q): What are the regularly spaced squares of SAT, SST, precipitation, TOA radiation at the poles, or is this a plotting artefact? (g)-(i) What do the values represent? The colorbar indicates that the scale can exceed -1 to +1. Is the ice cover annual mean, winter maximum, or something else?

P12 L10: "increased to"

Table 4: Are the uncertainties 1 standard deviation?

P13 L8: Remove brackets around (2020)

P13 L26: From a climate perspective, the volume transport of the AMOC is not important, it is the heat that it transports that matters. The heat diffusion described earlier in effect represents the ocean circulation.

P14 L11: less extensive

P14 L13-16: Do I understand correctly here that if you prescribe monthly pre-industrial climatological sea ice and run a pre-industrial simulation the AMOC collapses?

P14 L24: Whilst 0.58 is a reasonable albedo for bare sea-ice, it is an extremely low value for 'average' sea-ice, which will be covered by snow and melt ponds. An albedo closer to 0.75-0.8 would be more representative of typical ice cover.

P14 L27: Did you only vary one parameter in this set of experiments? It would be nice to see further detail of these.

P15 L8: consistently > consistent

P15 L17: "The first and the second coefficient are those used for the Northern and the Southern Hemispheres in the PlaSim-ML (T21) configuration." What does this sentence mean?

P15 L18: "We can notice..." I am not convinced that LSG could be said to have a slope similar to CESM. Between 1x and 1.5x CO2 the gradient of the line is actually negative.

P16 L20: typo: "eexoplanetary"

References:

Murphy JM, Sexton DMH, Barnett DN, Jones GS, Webb MJ, Collins M, Stainforth DA (2004) Quantification of modelling uncertainties in a large ensemble of climate change

simulations. Nature 430:768–772

Shi, Y., Gong, W., Duan, Q. et al. How parameter specification of an Earth system model of intermediate complexity influences its climate simulations. Prog Earth Planet Sci 6, 46 (2019). https://doi.org/10.1186/s40645-019-0294-x

Stainforth, D., Aina, T., Christensen, C. et al. Uncertainty in predictions of the climate response to rising levels of greenhouse gases. Nature 433, 403–406 (2005). https://doi.org/10.1038/nature03301

Williamson D, Goldstein M, Allison L, Blaker A, Challenor P, Jackson L, Yamazaki K (2013) History matching for exploring and reducing climate model parameter space using observations and a large perturbed physics ensemble. Clim Dyn 41:1703–1729

Williamson, D., Blaker, A. T., Hampton, C., and Salter, J. (2015), "Identifying and Removing Structural Biases in Climate Models With History Matching," Climate Dynamics, 45, 1299–1324.

Danabasoglu, G., & Gent, P. R. (2009). Equilibrium Climate Sensitivity: Is It Accurate to Use a Slab Ocean Model?, Journal of Climate, 22(9), 2494-2499.

Stouffer, R. J., & Manabe, S. (1999). Response of a Coupled Ocean–Atmosphere Model to Increasing Atmospheric Carbon Dioxide: Sensitivity to the Rate of Increase, Journal of Climate, 12(8), 2224-2237.

Shell, K. M. (2013). Consistent Differences in Climate Feedbacks between Atmosphere–Ocean GCMs and Atmospheric GCMs with Slab-Ocean Models, Journal of Climate, 26(12), 4264-4281.

Spelman, M. J., and S. Manabe, 1984: Influence of oceanic heat transport upon the sensitivity of a model climate. J. Geophys. Res.,89 (C1), 571–586.

---

## Referee Comment (RC2) · Anonymous Referee #2 · 8 Mar 2021

This manuscript presents a study where a simplified atmospheric model, PlaSim, well referenced and already tuned, is coupled with two different simplified ocean models in order to develop a new EMIC. A few parameters of the ocean models are adjusted so that the EMIC correctly simulates the current climate. $CO_2$ increase experiments are then performed and the ECS values are discussed, as well as the difference obtained with the two ocean models. The influence of ocean heat transport on sea ice feedback is shown.

I had a hard time finding interest in this manuscript primarily because I don't see its purpose. Is it an evaluation of an EMIC? Is it to isolate some fundamental processes

to study them in detail? Is it to present a tuning of the model? All these objectives are more or less present, but no question is really addressed, no subject is really deepened, and the bibliographical work is almost absent.

Finally, analysing the ECS as an emergent property for this kind of model no longer seems very relevant given that we know that some approximations that are made have a strong influence on climate sensitivity. It would seem to me more relevant to adjust the parameters of EMIC both on the current climate and on climate sensitivity. And this adjustment could not only concern the ocean but also the atmosphere and sea ice given the importance of atmosphere-ocean-sea ice interactions, an importance confirmed by the authors.

This manuscript looks more like the presentation of a work in progress than a mature work, and I consider that it needs to be extensively modified before it can be published in GMD.
* * *

---

## Author Comment (AC1) · 4 Apr 2021

We would like to thank reviewer #1 for his/her constructive comments which will help us to improve the paper and we would like to make some clarifications. Actually, the aim of this paper is not to perform an atmospheric tuning of PlaSim (which, as the reviewer points out and as we report in the paper, has already been the subject of an earlier study) but to focus on oceanic parameters in the currently available coupled versions of the original model. The paper uses the best configurations to explore equilibrium climate sensitivity in PlaSim and to discuss the most relevant feedbacks. These are results which, to our knowledge, have not been reported for coupled configurations of

PlaSim so far. For the mixed-layer ocean we focused indeed on the horizontal diffusion coefficient, as pointed out by the reviewer, since this is the main parameter controlling meridional heat transport. The range of different horizontal diffusion coefficients tested, from 10ˆ3 to 10ˆ6 mˆ2/s, was found to be sufficient to obtain a significantly improved model climatology. We discuss and explain our decision to implement a differential diffusion coefficient between the two hemispheres in the text. Actually, we did not test only three profiles of the vertical diffusion in the Large-Scale Geostrophic ocean as suggested by the reviewer: we show the most relevant profiles in the manuscript but we explored a wide range of profiles, with different diffusivities both at the surface and at the bottom of the ocean, as stated at page 8 line 11. The overall climatology of PlaSim has already been explored in the past. The purpose of the sections reporting model climatology and energy balances is to document the main properties of the climatology obtained in the new coupled model configurations after changing the oceanic parameters, but not to repeat an in-depth analysis of the model climatology and we decided to make a compromise in terms of complexity for these sections. We thank very much the reviewer for suggesting additional literature to be included in the discussion of the ECS experiments and for the useful detailed suggestions/typo corrections which we will implement. Following the reviewer's comments we have come to the conclusion that our paper may not fit well to the scope of GMD. For this reason, while we will use the suggestions by reviewer #1 to improve our manuscript (and we thank the reviewer for the time spent), we prefer to withdraw this submission from GMD.